# The 3Ps (Profit, Planet, and People) of Sustainability amidst Climate Change: A South African Grape and Wine Perspective

**Omamuyovwi Gbejewoh** [1] , **Saskia Keesstra** [2,3] **and Erna Blancquaert** [1,*]

1      Department of Viticulture and Oenology, South African Grape and Wine Research Institute, Stellenbosch University, Private Bag X1, Matieland 7602, South Africa; 22338225@sun.ac.za
2      Soil, Water and Land Use Team, Wageningen University and Research, P.O. Box 17, 6700 AA Wageningen, The Netherlands; saskia.keesstra@wur.nl
3      Department of Civil, Surveying and Environmental Engineering, The University of Newcastle, Callaghan 2308, Australia
*      Correspondence: ewitbooi@sun.ac.za

**Abstract:** Conventional agriculture has made the search for sustainability urgent, more so with regards to climate change. This has extended to the grape and wine industry, an important industry in South Africa in terms of labor employment and foreign exchange. This paper aims to review the current state of knowledge with regards to the three pillars of sustainability and with regards to climate change. In order to understand sustainability in South Africa, a historical context is needed, because the welfare of farm workers still retains vestiges of past Apartheid. Ecological responsibility and higher profits are the main reasons for sustainable practices. Additionally, water use, chemical use, and soil erosion are important environmental sustainability concerns. With regards to climate change, in terms of economic sustainability, there will be winners and losers and social sustainability issues will intensify as changes occur in farms. Table grape producers are relatively more profitable than wine grape producers. Furthermore, pest, disease, irrigation pressure will worsen as the climate warms. However, there are long- and short-term adaptation strategies such as changes in viticulture practices and grape cultivars, respectively, to stem the effects of climate change, but this may be stymied by cost and farmers' perceptions of climate change.

**Keywords:** sustainability; dimensions; global change; South Africa; table grape; wine grape





## 1. Introduction

There has been an increased contemporary awareness about the environmental impacts of agricultural production and consumption; since the 1960s, agriculture has relied largely on synthetic chemicals (fertilizers, herbicides, and pesticides) and mechanization to achieve increased levels of production at the least possible cost [1,2]. This period, known as the "green revolution", while it increased food production, brought detrimental consequences to the world's natural resources [3]. Consequently, sustainability and sustainable development from the Brundtland Report "Our Common Future" [4] and the 1992 Rio Conference on sustainable development was placed at the center of international, national, and regional agendas [5]. Presently, there are many policies, agendas, and strategies that aim to transition to sustainable development at different levels for general or specific levels of activities, from the United Nations Sustainable Development Goals [6], to the European Union Green Deal [7], to African Union's "Agenda 2063—The Africa we want" [8], to various national and regional policy agendas.

Sustainability has become a very important word in the world today. However, a single universal definition has so far been out of reach [9]. One of the first definitions was provided by the United Nations as it formed the World Commission for Environment and Development (WCED). Their definition was: "sustainable development is the development that meets the needs of the present without compromising the ability of future generations to meet

their own needs" [4]. However, since then different definitions have emerged, but it has since been a multidimensional concept built upon economic, environmental, and social principles [10,11] or the "triple bottom line" approach [12]. In the vision of the Sustainable Development Goals of the United Nations, true sustainability needs to address sustainability in the bio-physical environment, but also in the socio-economic environment. Solutions must be found in combining the needs for all three domains: biosphere, society, and economy [13].

The calls for sustainability have never been greater in the agro-food industry to address the environmental impacts and resource inefficiencies of the current system [14]. This call has extended to all sub-sectors of the industry, even the grape and wine industry that traditionally has not been viewed as a particularly environmentally inefficient industry [15]. Regardless, sustainability has been of great concern for the grape and wine industry particularly because of the risks associated with climate change. Numerous authors have reported on the importance of climate in grapevine physiology, growth (phenology), yield, and the subsequent fruit and wine quality [16–23].

In South Africa, sustainability and climate change are especially important concepts to the grape and wine industries because they are major contributors to the South African economy. South Africa is the seventh largest table grape exporter, commanding 6.2% of the export market share and employing almost 80,000 permanent and seasonal workers [24]. In terms of wine production, South Africa is the ninth largest wine producer (3.3% of world production), and sixth largest exporter of wine (4.9% of world exports) [25].

Considerable research has been conducted on the three individual pillars of sustainability and in the context of climate change in the South African grape and wine industry [25–30]. However, a major gap in these studies is the provision of a holistic overview of the three pillars in tandem. Consequently, this paper aims to review the state of current knowledge concerning the three pillars of sustainability in the grape and wine industry in South Africa in the context of climate change. In this review, the objective is to:

(i)　analyze why sustainability is important to grape and wine farmers;
(ii)　analyze current trends in the economic, environmental, and social sustainability of grape and wine production and how climate change is affecting these trends.

The framework for the analysis for the current trends in sustainability will be to discuss each pillar (economic, environmental, and social) of sustainability separately at first as a standalone concept. Thereafter, climate change will be introduced into these pillars; thus, the effects of climate change in each pillar (economic, environmental, and social) of sustainability will be further discussed separately. The outline of the paper is as follows: first, a description of the systematic review process; next is an explanation of the results of the selected papers from the review process. Thereafter, a discussion of why sustainability is important to grape and wine farmers and a historical context of grape and wine production in South Africa is provided to better understand sustainability trends in the country. Afterwards, a description of the economic, environmental, and social trends of grape and wine production in South Africa in the context of climate change. Finally, climate change adaptation strategies are discussed and areas where research is lacking and in need of further development is given.

To the best of our knowledge, this is the first review paper of its kind that focuses on all three sustainability pillars simultaneously in the context of climate change in the South African grape and wine industries.

## 2. Methodology

This review followed the guidelines set by PRISMA [31] for a structured review as shown in Figure 1. The review used a mixed-method approach which included quantitative and qualitative research. Web of Science and Scopus was used between April 2020 and June 2020 to obtain journal papers and conference proceedings. The search string words in Web of Science and Scopus Database were: TITLE-ABS-KEY (("sustainability*" OR "sustainability pillar*" OR "climate change*") AND ("viticulture*" OR "vineyards*" OR "wine*" OR "grape*")) There were no temporal limitations for this study. The papers

were downloaded and exported to Mendeley Desktop where duplicates were immediately removed. The inclusion criteria were theoretical papers, and qualitative and quantitative studies. Book chapters, papers not in English, and conference proceedings were not considered for this review. The article titles and abstracts were screened, and papers not related to agriculture and parts of viticulture and winemaking deemed not relevant (e.g., wine chemistry, flavor chemistry, wine aroma, sensory evaluation, grapevine biology, wine microbiology, etc.) were removed. Furthermore, whole texts were analyzed and papers that dealt with other aspects of agriculture (e.g., crop and animal production) were removed except if they dealt explicitly in sustainability and climate change.

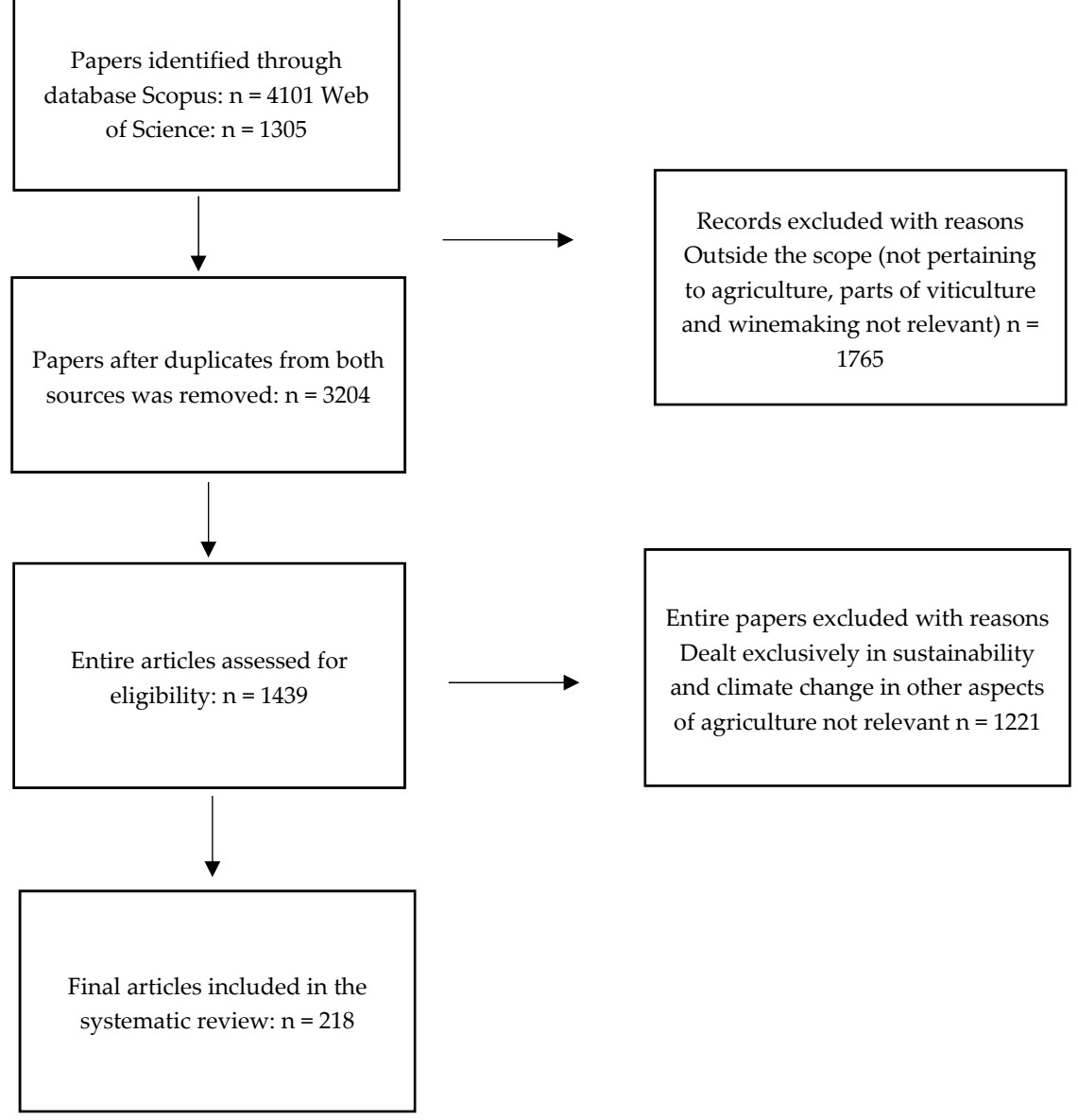

**Figure 1.** A PRISMA flowchart of the PRISMA systematic review process.

### 3. Results

An initial search of Scopus and Web of Science database yielded 4101 and 1305 papers, respectively. After duplicates were removed, this was reduced to 3204. Thereafter, after article titles and abstracts were examined, 1765 articles were excluded according to the aforementioned reasons, which reduced the number of articles to 1439. After that,

the entire papers were examined and whole papers were removed according to the reasons given above. This gave a final paper count of 218. According to Figure 2, the majority of the selected papers were focused on the pillar of environmental sustainability (47.3%). This was followed by the pillar of economic sustainability (20.5%) and the pillar of social sustainability (13.2%). The number of research papers that dealt with all three pillars simultaneously was low (3.9%).

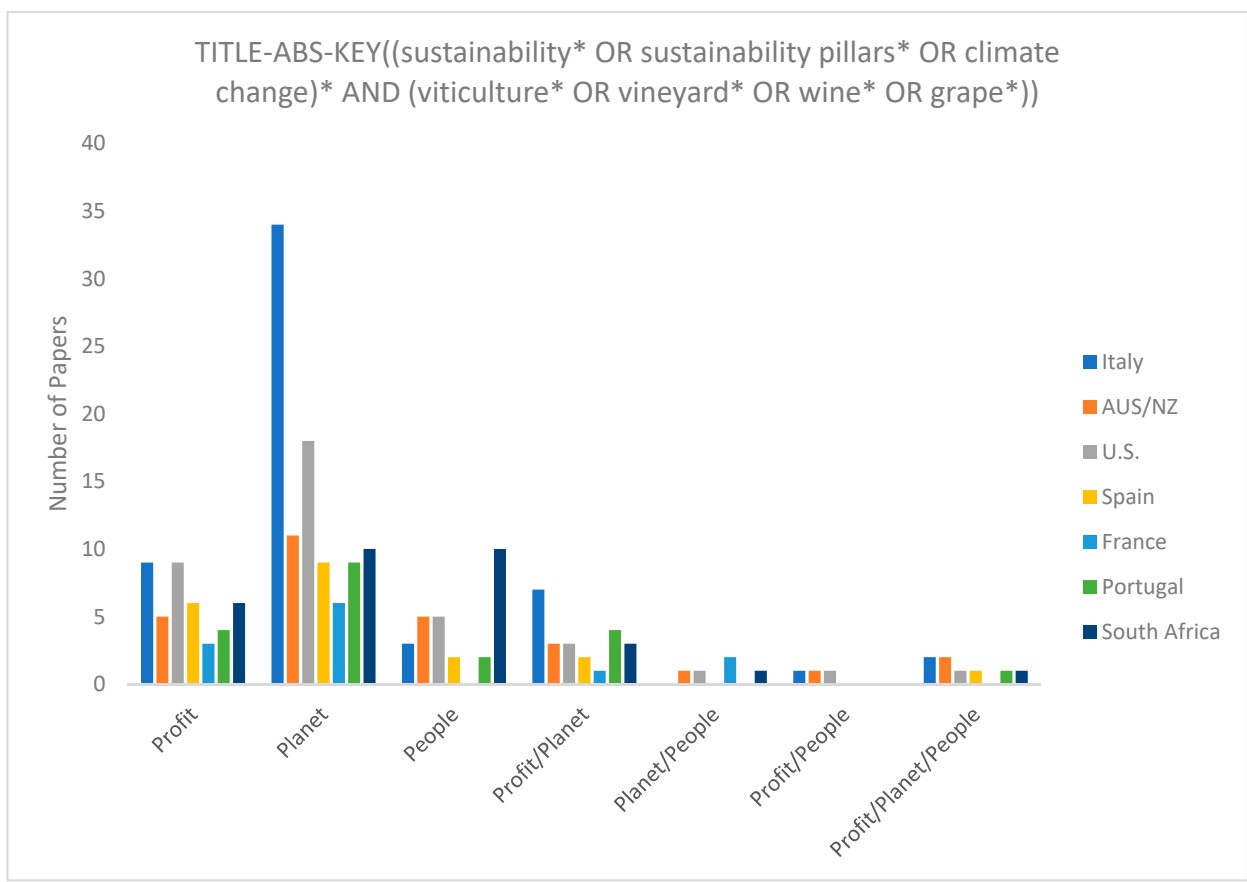

**Figure 2.** Bar graph showing the number of papers that dealt with the pillars of sustainability.

An overwhelming majority of the studies were conducted in Europe (Italy, France, Portugal, and Spain) followed by the United States, Australia, and New Zealand. Research in South America, Asia and Africa were abysmally low. The studies were published in a diverse range of journals, ranging from the *Journal of Cleaner Production*, *Sustainability*, *Journal of Wine Research*, *Journal of Wine Economics*, etc.

## 4. Discussion

### 4.1. Why Do Grape Farmers Become Sustainable?

Motivations for sustainability usually fall under ethical/personal/ecological responsibility, operation efficiency, marketing positioning/competitiveness, legitimacy/regulatory compliance, product quality/differentiation, higher profits, stakeholder pressure, and consumer demand [32–38]. Furthermore, business age, size, and ownership are factors that also play a role in the adoption of sustainable practices [39–41]. Hamman et al. [36] and other authors [42,43] found that in South Africa environmental responsibility is the major driver for sustainable practices and that legitimacy and competitiveness play a minor role. However, they emphasized that most sustainably proactive farms are characterized by environmental responsibility and a possible competitive edge. It was also reported that small- and medium-scale enterprises and family-owned businesses are more environmen-

tally proactive, because managers can translate their personal environmental beliefs to organizational practices due to the high degree of control on operations [44,45] Finally, potential barriers to sustainability practices may include cost, time intensity, lack of information, abuse of the sustainability concept ("greenwashing") or a perception of how a good, well-maintained farm should look (clean and without weeds) [9,32,38,46]. Regardless, the adoption of sustainability practices usually depends on whether the perceived benefits outweigh the cost [47,48].

### 4.2. The Historical Context of Grape and Wine Production in South Africa

For a better understanding about sustainability in the grape and wine industry in South Africa, the historical and political context is important. Between 1917 and the mid-1990s, the regulatory system in the South Africa wine industry was presided over by the Koöperatieve Wijnbouwers Vereniging van Suid-Afrika (KWV), who instituted planting quotas, minimum prices, and methods of surplus removal, and were the sole exporter of wine. Wine production was dominated by co-operative cellars who pooled resources to sell grapes in bulk and farmers were paid according to tonnage delivered. These co-operatives were closely linked to the network of white power in the Western Cape, because rural civil society in the province was dominated by the white landed settler elites [49]. These co-operatives encouraged mass production and rewarded growers who could deliver high volumes of low-quality grapes (high sugar levels, unbalanced acids, pH, and low phenolic content—key determinants of wine quality). This orientation coupled with the imposition of international trade sanctions because of Apartheid policies in the country brought the industry to a halt, although it consequently survived through domestic consumption and exports of low-quality wine to Eastern Europe [50,51]. This mass production of grapes was dependent on cheap black labor which, until the 1980s, was characterized by racial hierarchy and authoritarian paternalism adapted from the earlier Cape slave society [52]. White settler elites controlled most of the commercial farming in the Western Cape and beyond, with values of white patriarchal mastery that shaped the relationship between farm owners and farm workers. Even with attempts in the 1980s to "modernize" labor relations (as a result of pressure on Apartheid policies) with workers' education and skill development, and even, ironically, research into fetal alcohol syndrome (which was largely caused by the "dop" or "tot" system), this notion of white mastery did not change but instead created a kind of "neo-paternalism" [53–55].

With the political transition of the early 1990s and a change in the economic and political power that had previously benefited the white elites, a slew of labor and employment legislation ranging from basic labor laws to minimum wage was passed to limit the control by farmers of workers' lives. Even though labor laws have significantly weakened the paternalist labor, it has not decisively transformed it; the state is most often too far away to enforce their laws, and farmworkers are reticent to fight for their rights because maintaining close and cordial relationships with farm owners are just as important [56].

With the lifting of trade sanctions and opening of the export markets following the political transitions of the 1990s, South African wine was thrust into an international market that was going through a lot of changes. Firstly, the global economic downturn was putting pressure on the global beverage industry, and global wine consumption was decreasing. Secondly, supermarkets were growing in importance as wine retailers which changed how wine was consumed and marketed, and lastly, was the increasing prominence of premium and super-premium branded wines and the falling prospects and consumption rates of low price, blended, and bulk wines which hitherto the country was focused on. All these had contradictory implications for producers. Although new markets meant new opportunities, these supermarkets had stringent purchasing requirements through strict phytosanitary, technical, and ethical requirements. Furthermore, deregulation and globalization meant an oversupply of wine coupled with competition within the country but also with other wine-producing countries for much sought-after supermarket contracts, placing producers at a disadvantage when bargaining with wine retailers [56]

For table grape producers, just like wine, supermarkets were becoming the dominant retailers of fresh produce. These supermarkets no longer purchased fruits in the open market but through integrated global value chains (GVCs). Supermarkets usually work with a close group of agents in the value chain to plan and preprogram their requirements annually to meet changing consumer needs. Their dominant position in the value chain allowed them to influence their agents and suppliers and exert increasing pressures on fruit growers to meet tight—albeit flexible—production schedules, and comply with quality, environmental and social standards. However, these supermarkets rarely have written contracts with suppliers that provide a guarantee of purchase, except for a verbal agreement. Furthermore, their purchase of fresh produce is mainly on a "consignment" basis, where prices are not agreed until very close to the point of final delivery. Additionally, even though they demand and dictate standards, the prices they pay are subject to the forces of demand and supply on the open market. Like wine, globalization and deregulation following the transition to a democratic government led to the dismantling of Unifruco, the single export channel of fruits. This resulted in increased competition within South Africa and between other exporting countries such as Chile, which exports fruits within the same "export window" as South Africa, leading to an oversupply and a subsequent decrease in prices [57].

Grape and wine producers consequently responded in ways that were still in their control, through the contraction, casualization, and externalization of labor [58,59]. However, it is important to note that this trend towards flexible employment is not unique to South Africa; research of the literature has emphasized the same trend worldwide, especially in the agricultural sectors of developing countries [60]. This is essential to understand the world that South African grape and wine producers entered in the early 1990s, and it is important to view any of the sustainability pillars through this lens.

### 4.3. Climate Change in Grape and Wine Production

Climate change is expected to impact viticulture through an increase in air temperature and a shift of the ripening period towards earlier and usually warmer parts of the season [61]. Mean temperatures for traditional viticultural zones have increased by 1.7 °C between 1950 and 2004 [62], and changes in grapevine growth and development have already been found [63–66], influencing grape yield and berry and wine quality [62,67] through a decrease in berry acidity [68,69] and increase in sugar content [63]. Furthermore, changes in temperature and rainfall patterns [70] may significantly modify viticultural zones in Europe [71,72] through severe dryness [73], organoleptic and organic acid degradation [20,64], and high potential alcohol content [74], although less modification is expected for South Africa [26,62]. Increased temperatures may also open new viticultural zones that previously were unsuitable for grape production in Northern and Central Europe [75,76], Western North America [77], and cooler and higher altitude regions in the Western Cape of South Africa [29,78].

### 4.4. Economic Sustainability of Grape and Wine Production

Economic sustainability in its simplest term means how farms in business stay in business. Economic sustainability is intricately linked to environmental and social pillars. Consequently, while only good economic performance might be beneficial in the short term, it is not necessarily so in the long term because neglecting the environmental and social dimensions may be a barrier to long term survival. Thus, effectively managing the environmental and social dimensions of businesses can also make farms economically sustainable [79]. Economic sustainability is usually viewed as economic viability, which means whether the farm can survive in the long term in changing economic contexts. These changes in economic contexts may be driven by changes in inputs and output prices, yields, governmental regulation, while the long term implies the entire working life of the farmer or even the working life of subsequent generations of successors of the farm. Economic viability is usually measured through profitability, stability, liquidity, and productivity [80].

However, economic sustainability sometimes extends beyond these indicators to others such as autonomy in various forms, examples are financial autonomy (less pressure from debts), diversification of income, and autonomy from subsidies [81].

Following the lifting of trade sanctions, South Africa's grape and wine exports increased four-fold between 1994 and 2004 but it has since plateaued. Additionally, tourism-related activities (tours, restaurants) have been an important source of income for South Africa's vineyards and cellars. However, South Africa's grape and wine farmers have fared better than other sectors in the agricultural economy [82]. For the last 10 years, South African wine farms have averaged a net farm income (NFI) that is less than what is required for sustainable grape production, but the situation is gradually improving. For example, in the 2018 vintage year, vineyards averaged an NFI of ZAR 14,957/ha compared to the ZAR 30,000/ha required for sustainable grape production. Moreover, for the 2019 vintage year, vineyards averaged ZAR 20,617/ha compared to the ZAR 34,000/ha required for sustainable production. However, these increases have been driven largely by yield increases, and over the past two years, this has been coupled with rising grape prices. While this is remarkable, it is also unsettling, given that future yield decreases are expected given the increasing percentage of aging and older vineyards. In fact, according to Figure 3, for the first time in 16 years, vineyards that were younger than three years old made up less than 10% of total hectares, and vineyards older than 20 years constituted over 20% of total hectares, which is contrary to the general knowledge that these figures should be 15% or more for vineyards aged three years or younger, and less than 15% for vineyards aged 20 years or older.

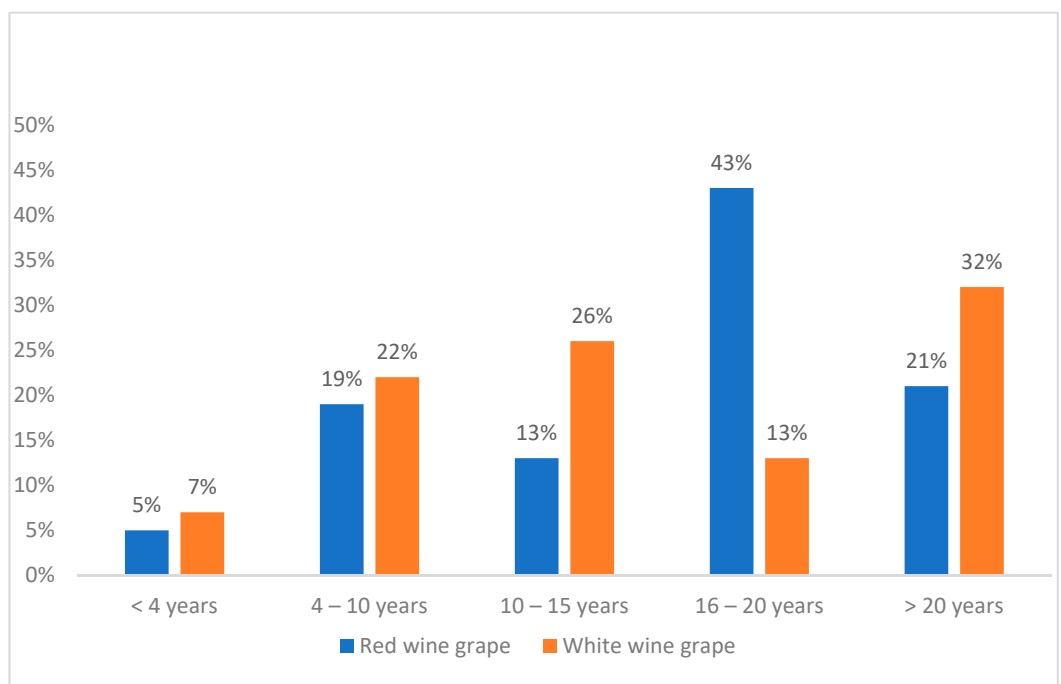

**Figure 3.** Age distribution of vineyards in South Africa [83].

Consequently, future increases in gross income, NFI, and profitability will largely need to come from a further increase in grape prices. However, it should be noted that the percentage of profitable vineyards (NFI > ZAR 34,000/ha) increased from 15% to 28% between 2015 and 2019, while the percentage of unprofitable vineyards decreased from 40% to 30% between 2016 and 2019 after an increase from 30% to 40% between 2015 and 2016; however, the majority of wine farms (40% in 2019) are still barely profitable, with an NFI between ZAR 20,000–34,000/ha [83]. Figures 4 and 5 show the increasing production

costs of vineyards and the relatively modest profitability of vineyards in South Africa over the years.

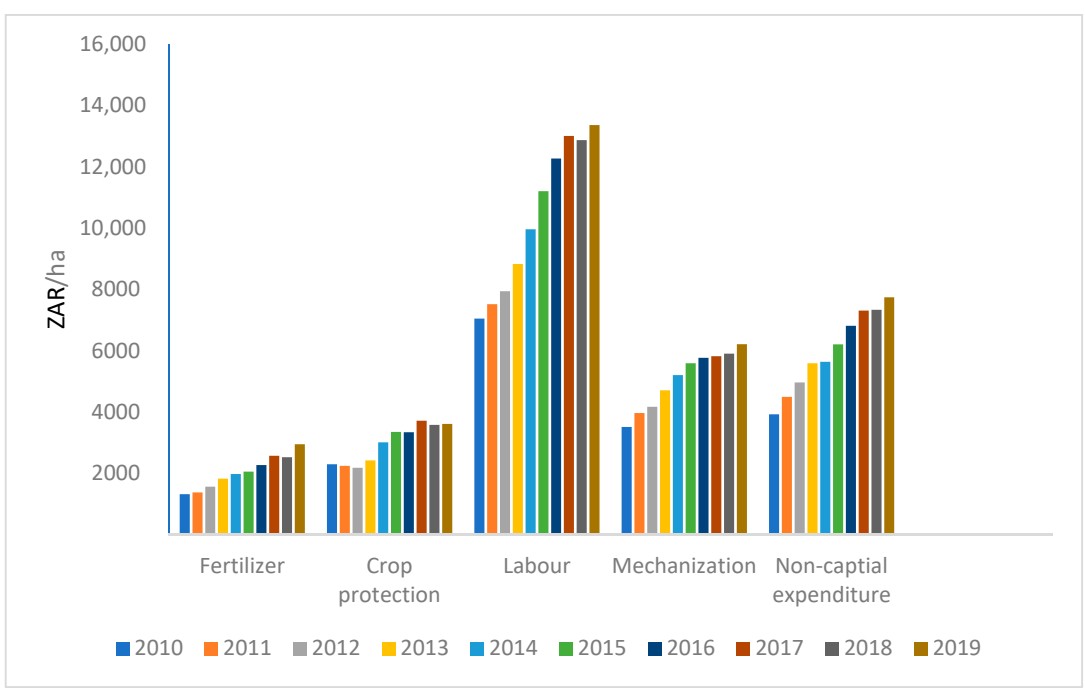

**Figure 4.** Production costs of vineyards [83].

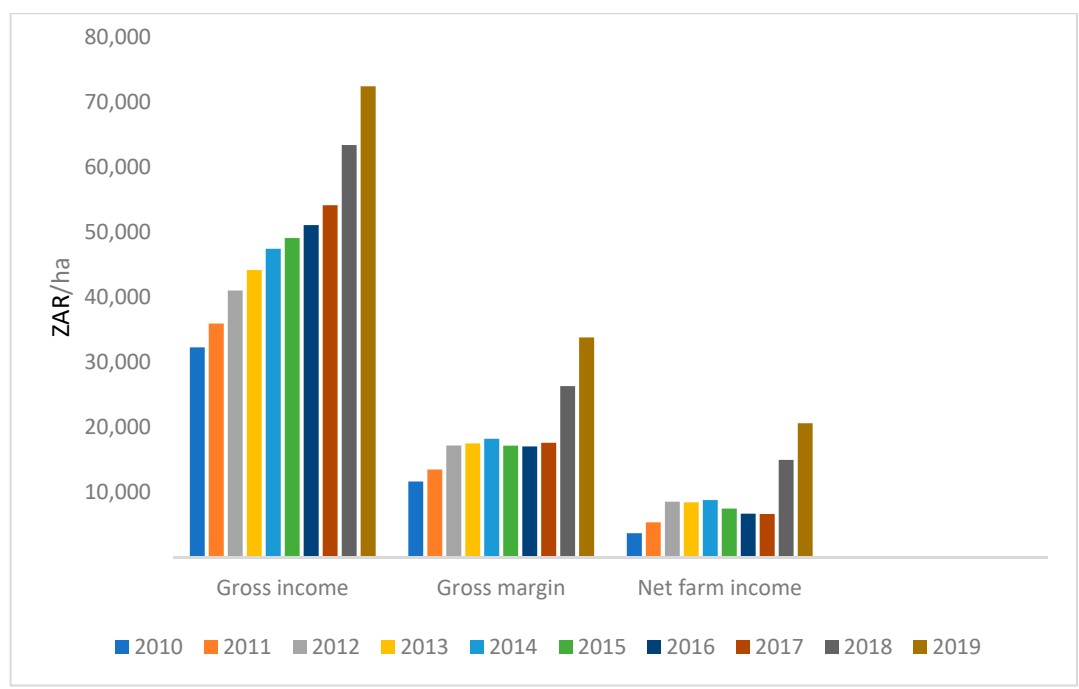

**Figure 5.** Profitability of vineyards on average [83].

For table grapes, the NFI income projection is roughly ZAR 46,000/ha. However, even though a projected increase in NFI is expected for table grapes, according to Table 1, the export outlook is worrying because of its dependence on the E.U. and U.K. markets. This is because even though production is expected to increase in South Africa, the population growth of the E.U. and the U.K. is worryingly slow and, in some countries, even negative,

meaning that consumer demand will probably remain at current levels. Consequently, if market access is not expanded, further oversupply will be detrimental to the price of table grapes in the long term. Canada is an emerging export market to which table grape producers are looking to export, but there is also competition from other table grape producing countries looking to export there [84]. China is also another export market where South African table grape producers are looking to increase their footprint, because it has a growing economy and population with strong cultural importance for fruit consumption, complementary growing seasons, and both governments have pledged to increase bilateral trade [85].

**Table 1.** Regional table grape export market split 2019/2020 (4.5 kg Equivalent Cartons) [84].

| Region | Export Market |
| --- | --- |
| European Union | 31,400,602 |
| United Kingdom | 15,793,685 |
| Canada | 4,221,802 |
| Far East | 2,951, 997 |
| Middle East | 2,902,807 |
| South East Asia | 2,877,238 |
| Russian Federation | 1,193,984 |
| Africa | 877,039 |
| United States | 488,003 |
| Indian Ocean Islands | 275,262 |
| Other | 190,457 |

*4.5. Economic Sustainability in Climate Change of Grape and Wine Production*

The economic consequence of climate change on grape and wine as shown in Table 2 is generally hard to predict because variability is large, and the climate change process is non-linear [86]. Nemani et al. [87] showed that the increased temperature associated with climate change was an advantage to the wine industry in California, because frost occurrence reduced by 20 days and the frost-free period increased by 65 days. Adams et al. [88] corroborated these results, with a 90% and 65% increase in yield with and without $CO_2$ fertilization, respectively. However, a decreased yield for table grapes and relatively stable yields for wine grapes with increased temperature have also been shown [89]. Regardless, continued global warming may turn any possible gains into definite losses [90].

Higher yields are usually correlated with reduced wine quality [91]. However, even though this negative relationship between yield and quality is true most of the time [91], it is not always the case [92,93]. Furthermore, wine quality is also related to alcohol content, because higher temperature produces sweeter and stronger wines [91]. Even though many viticultural regions have been trending towards higher alcohol content, climate change is not fully responsible for this trend; consumer preferences and viticultural practices also play significant roles [93,94]. Wine quality has also been associated with higher prices. Alston et al. [95] showed a 61.6% increase in wine prices with a corresponding 1 °C increase in growing season temperature in Bordeaux. Similar results were found by Jones and Storchmann [96] and Chevet et al. [97]. These results may hold for cooler regions [91] but for warmer regions, there is a maximum peak in prices with regards to increased temperature, above which further increases in temperature reduces prices [98,99]. Based on current literature, evidence shows that there will be both winners and losers from climate change [91].

**Table 2.** Interaction of key economic sustainability indicators with temperature.

| Economic Indicator | Temperature |
|---|---|
| Yield | 90% and 65% increase in vine yield with and without $CO_2$ fertilization, respectively, corresponded with 3 °C increase in temperature [88] |
| Wine quality | 0.23% increase in Brix levels per year between 1980 and 2005 [91] |
| Revenue | 150–180% increase in revenue with a 3 °C increase in temperature [91] |
| Price | 61.6% increase in price with 1 °C increase in temperature [95] |

*4.6. Environmental Sustainability of Grape and Wine Production*

Grape and wine production have been subject to less regulation compared to other industries such as the manufacturing, chemical, and mining industries [33,100]. This is probably due to the preconception of grape and wine production as environmentally safe [101]. The most important issues related to environmental sustainability as shown in Table 3 are water use efficiency, use of chemical crop protection, and soil erosion.

**Table 3.** Environmental sustainability concerns of grape and wine production [15].

| Environmental Indicators | Environmental Concerns |
|---|---|
| Water use | Inordinate water use coupled with inaccurate and/or absent data on water use |
| Organic and inorganic waste | Lack of data on waste generated coupled with limited and/or absent recycling programs |
| Synthetic chemicals use | Excessive use of synthetic chemicals with absent data on chemical use |
| Energy use and greenhouse gas emissions | Energy use in addition to $CO_2$ generated is an often-ignored environmental concern. |
| Ecosystem impacts | Soil erosion, destruction of local habitats, loss of biodiversity associated with vineyard monocultures, local pollution and contamination, and competition for water resources with other aspects of agricultural production |

4.6.1. Water Use Efficiency

Water is a very important resource for grape and wine production, for which usage in viticulture and winemaking can vary according to the location and size of the farm [37]. For example, water footprint can provide important information on the water use of a specific portion of the farm, and strategies can be developed from this information to improve the water use efficiency. Water use is broadly categorized as blue, green, and grey water for agricultural use [102]. Generally, water use can be categorized as direct or indirect. Direct application of water refers to the application of irrigation, fertilizers, and herbicides, while indirect water use includes water use for agrochemical dilution [102]. In wine production, direct use of water in the cellar includes the washing of equipment (before and after crushing), winemaking, cold stabilization, and sanitation, while indirect water use in the cellar includes water for chemical dilution and water for waste removal [102]. Jarmain [102] reported that table grape and wine production in South Africa showed that on average, table grapes used 619 L of water for every 4.5 kg carton (industry standard) of table grapes produced; on average, 647 L of water was used for every 750 mL of wine produced [102]. Water use can consequently have important effects on the quantity and quality of water resources. Evidence suggests that vineyard and winery managers do not know or keep data on the quantity of water used and/or wastewater generated in their organization [15]. In a South African study, 80% of wine farmers could not accurately give their water use and even underreported the exact value by as much as 60% [103]. An Australian study showed that about 5% wine farmers still used over 8 L of water to

produce a bottle of wine, regardless of the fact that research on the best management practices has reported the use of 0.4 L of water [104]. Concerns with the excessive use of water in viticulture and winemaking include the contamination of surface and groundwater sources, and the inappropriate disposal of wastewater [105,106]. Practices such as the use of drip line irrigation and partial root drying have been championed [19,107], and even reduced water use has been shown to be of benefit to wineries; a Canadian study showed a 6% increase in grape yield with a 30% reduction in water use [108], while a South African study stressed the importance of remote sensing and earth observations technology for quantifying water use over large areas [102].

### 4.6.2. Organic and Inorganic Waste

Organic and inorganic solid waste are unavoidable consequences of grape and wine production [15] and is one of the most important environmental concerns facing the industry [105,109]. Furthermore, just like water, there are a lack of data collected by farms [106]. Organic waste includes winery effluents such as grape marc, lees, and pomace stalk, some of which have the potential for reuse while others are of practically no value [110,111]. Inorganic waste, on the other hand, includes packaging materials and used chemical containers [37]. Landfills and incinerators are popular options for organic and inorganic waste disposal, and even though there is a growing market for organic waste and success with recycling programs, there is still room for further improvement [112,113].

### 4.6.3. Chemical Use

Similarly to other agricultural sectors, chemical use in vineyards includes fertilizers, pesticides, and herbicides, and in some countries chemically-treated timber is used for vineyard trellising [114,115]. Chemical use in wineries includes chemicals for cleaning operations, sanitation, and wine preservation [106,113]. The chemical use in vineyards is especially disconcerting; it has been shown that although European vineyards occupy only 3% of cropland, they use 15% of all synthetic pesticide applications [116]. Furthermore, just like other agricultural sectors, chemical use in vineyards is associated with contaminated run-off, spray drift, reduced soil fertility, reduced bee populations, damage of vineyards' natural defense networks, while chemical use in wineries affects the quality of wastewater making effective treatment before disposal cumbersome [117].

### 4.6.4. (Un)Sustainable Agronomic Management and Resulting Soil Loss

Soil erosion is an environmental risk that is particularly severe in vineyards because of soil tillage, poor organic matter content, and climatic conditions [118,119]. Consequently, this leads to a loss of soil fertility, soil quality, and loss of ecosystem services [120]. It should be noted, however, that extensive soil loss is not limited to vineyards; different authors have reported similar problems in various other crops [121–124]. Research has suggested that soil loss in vineyards is above the level that amounts to tolerable soil loss, less so for older vineyards with more organic matter content and higher bulk density in relation to younger vineyards [125] Moreover, accurately measuring soil loss in vineyards is fraught with difficulties, because different methodologies available tend to give different results. Thus, there is a need to improve the accuracy of measurements [126]. However, research has shown that there are various practices to mitigate the effects of soil erosion such as terracing, sediment fences, check dams, grass margins, contour farming, and the use of cover crops [127,128].

### *4.7. Environmental Sustainability in Climate Change of Grape and Wine Production*

In the context of climate change, there is concern that increased temperature associated with climate change will cause increased pest and disease pressure on crops [129–131], and these changes are already taking place [132,133]. Increased temperature may cause the increased survival of pests and diseases during warmer winters and may cause the range of pests and diseases in a region to change because pests may move to cooler regions that

were previously unsuitable for their development, as evidenced by the invasive spotted wing *Drosophilia* fruit fly, native to Southeast Asia, but has increasingly been spreading to Europe and the United States. [134,135]. Although pest movement to cooler regions is more likely due to globalization than climate change [136], their increased survival in these cooler regions is probably due to milder winters [135]. On the other hand, even though the increased temperature is likely to allow more pest generations in a growing season, as evidenced by the rice strip virus transmitted by the small brown planthopper [137,138], this may be offset by the early maturity and earlier harvest dates causing asynchrony and limiting pest damage [139]. However, results like this should be viewed with caution; pests may be able to maintain their synchrony with the target host [140] or adapt, and very likely restore, this synchrony [141].

The increased threat due to climate change is the increased erratic nature of the climate. Rainfall becomes more unpredictable and more intense when the climate warms. This causes higher soil erosion rates due to increased runoff in high intensity storms and higher soil detachment rates due to increased splash erosion [127]. Increased warming with associated increased evapotranspiration and increased frequency and intensity of extreme events such as droughts, wildfires floods, and heatwaves [142] will bring increased pressure for irrigation and less reliance on precipitation, more so for old world producers than new world producers [92]. For South Africa, this situation is especially dire because the country is one of the most water-scarce countries in the world, with large areas classified as arid or semi-arid [27].

Consequently, the expected increase in irrigation and water use in vineyards and wineries is likely to bring associated risks of erosion and silting of water bodies, especially as vineyards move uphill to areas of lower temperatures [30,42,106,143], salt build-up in soils which is detrimental to vines [93], and increased competition for water and land resources from other sectors of agriculture arguably deemed more important in terms of food production [15,144], and consequently pushing grape and wine production from traditional areas to more marginal areas with fewer resources [68]. Furthermore, the fynbos region of the Western Cape of South Africa where a significant portion of grapes are grown is fire-prone, and adaptation to frequent fires is a natural feature of the fynbos vegetation [145]. Even though studies are scarce, increased frequency of wildfires is expected with climate change [26,146] with effects of increased soil erosion after a fire [147].

Finally, the prospects of vineyard relocation with further warming are expected to bring biodiversity conservation concerns [77,148] especially in the Cape Floristic region of South Africa [78], one of the biodiversity hotspots in the world [149] and a major grape and wine producing region. Even though there are programs such as the Biodiversity and Wine Initiative (BWI) in South Africa to mitigate against the impacts of vineyard expansions and possible relocation through botanical audits, plans to preserve endangered and significant species, and setting aside land for biodiversity conservation [150,151] a large number of grape and wine farms are small- and medium-scale enterprises [152] without particularly large tracts of land; therefore, a majority of the reserved areas are likely to be small scattered fragments, making a formal reserve system particularly difficult [153,154].

### 4.8. Social Sustainability in Grape and Wine Production

The South African grape and wine production industries were infamous for some of the worst working conditions in Apartheid South Africa, and even though conditions improved following the political transition and the passing of legislation to improve workers welfare, transformation in the grape and wine industry still lags behind other sectors [150,155]. The casualization and externalization of labor especially after the transition to a democratic government was detrimental to farm workers. Research shows that almost three million farm workers were evicted from farms between 1950 and 2004 and then rehired as seasonal and casual workers, sometimes under worse conditions than before [156]. This change in labor structure has both pros and cons in terms of workers relationships with farm owners. On the one hand, while the "firm but generous" relationship,

accommodation and discounted goods and services afforded to workers were gone, these farm workers were also free to unionize more easily and fight for their rights [157].

The neoliberal economic policies of the government have made them reticent to interfere in the relationship between farm owners and farm workers, so much so that apart from the lack of manpower, the government are reluctant to enforce their own labor laws. For example, it is possible for farm owners to apply for exemption from minimum wage labor laws [158]. In cases where farm owners have had to comply with labor laws, researchers argue that this has accelerated the rate of casualization and externalization of labor [159]. For example, one study found that when minimum wage was introduced in 2003, farm workers' wages increased by 17% but agricultural employment decreased by 13% [160]. It has also been found that agricultural employment reduced by 8.3% as minimum wage was increased by 52% [161]. However, this research and arguments for less intervention in farm owners and farm worker relationships by the government belies the fact that since the opening of the export markets following the transition to democratic government, exports and the income in commercial farms have increased exponentially [158]. Farm workers has always been viewed as expendable, regardless of the economic situation of the farm owner, and that is not going to change anytime soon.

Although their research was limited to female farm workers in Western and Northern Cape, because women are more likely to be casualized and paid less, as shown in Table 4, Devereux [158] found that more than half (55%) were not aware of the sectoral determination that deductions from wages should be limited to 10% of wages; 40% had not signed an employment contract; for those that signed, more 80% of seasonal workers did not receive a copy of their contracts; 41% were paid below minimum wage, more so for those paid fortnightly and monthly and less so for those paid daily and weekly; and almost 80% of workers had had deductions from their wages (some legitimately, others less so). In addition, 63% of farm workers did not have access to bathroom facilities, 62% were compensated for injuries incurred on farms and about half (51%) of these injuries were reported to the Department of Employment and Labour, 66% of farm workers were not provided with protective clothing from pesticides when spraying, membership in unions was abysmally low at 12%, 73% of farm workers claimed that farm owners do not allow union reps on farms, and 28% claimed the farms had never been visited by labor inspectors. It should be noted that violations of workers' rights are not limited to Western and Northern Cape; similar patterns of violations have been recorded in Eastern Cape [162], Limpopo [163], North West [164], and Free State [165].

**Table 4.** Violations of workers' rights in vineyards and wineries [158].

| Social Indicator | Province | | Workers | |
|:---:|:---:|:---:|:---:|:---:|
| | Western Cape | Northern Cape | Permanent | Seasonal |
| Did not sign a contract | 29.4% | 54.2% | 23.9% | 52.4% |
| Received a copy of their contract | 16.2% | 60% | 37.2% | 17.5% |
| Paid minimum wage | 62.4% | 59.6% | 73.2% | 51.6% |
| No access to facilities | 57.2% | 71.1% | 52.2% | 72% |
| Compensation for injury incurred at work | 61.5% | 60% | 64.4% | 61.2% |
| Injury incurred at work reported to the labor department | 55.2% | 37.1% | 64.4% | 36.7% |
| No protective clothing at work | 52.7% | 74.3% | 54.5 | 73.3% |
| Exposed to pesticides | 45.3% | 95.8% | 63.5% | 69% |
| Trade union membership | 13.6% | 9.9% | 13.8% | 9.5% |
| Farm owner does not allow union reps on farms | 64.7% | 86.6% | 68.6% | 76.8% |
| Farm owner prohibits attending union meetings | 49.3% | 63.4% | 47.8% | 60.7% |

### 4.9. Social Sustainability in Climate Change of Grape and Wine Production

Social sustainability research has been few and far between, and research on social sustainability in climate change has been even more so. Grape and wine production has strong cultural, social, and historical ties to a viticultural zone, and the concept of terroir embodies

this [166]. Consequently, climate change will have different social consequences according to different contexts. In many old world viticultural zones, where terroir holds very strong meanings, changes in grapevine varieties, viticultural practices, and even possible vineyard relocation will affect regional, cultural, and social identities [166]. Furthermore, viticulture and winemaking are significant employers of labor in many viticultural zones and may be severely affected by changes in viticultural practices and vineyard relocation [167,168]. Additionally, the capacities of grape and wine farmers to adapt to climate change are influenced by social, economic, and political circumstances [169].

### 4.10. Climate Change Adaptation Strategies

Regardless of the various ongoing and expected effects of climate change in viticulture and winemaking, there are various short-term and long-term adaption strategies to reduce the effect of climate change in viticulture. Short-term adaptation strategies include viticultural practices to delay ripening [170], the use of sunscreen and shade nets to protect from sunburn and extreme heat [171,172], deficit irrigation practices as a water-saving measure and to take advantage of the relationship between vine–water status and yield [173], integrated pest management practices to adapt to the possibility of increased pest pressure, and soil management practices (conservation tillage, use of compost, mulches, cover crops) for soil and plant protection, carbon storage, and reducing greenhouse gases emissions [174]. Long term adaptation strategies include changes in training systems for higher water use efficiency, lower sugar accumulation, delay of the maturation period, selection of grape varieties and rootstock to those better adapted to the expected effects of climate change, genetic breeding for the development of climate change-tolerant varieties, and finally, usually as a last resort, vineyard relocation to cooler, higher altitude, higher elevation, coastal areas, and areas with lower solar radiation [174]. However, it should be noted that adaptation strategies that do not consider the economic, social, political, and cultural constraints at the farm, regional and national level are likely to be unsuccessful [168,175,176]. Furthermore, the decision to adopt an adaptation strategy will depend upon a farm or organization's capacity to change, the perception of their vulnerability to climate change relative to other risks, and the risks and opportunities associated with adaptation [177].

### 4.11. Knowledge Gaps and Future Research

This review has implicitly shown that an overwhelming majority of research in sustainability and sustainability in climate change has been conducted in Western countries. However, the historical context of the country presents a unique opportunity in sustainability research. Research in sustainability has essentially tackled one pillar at a time, and research in all three pillars is abysmally low. This needs to be remedied because grape and wine farmers battle all three pillars at the same time. Revenue from grape and wine production in South Africa has plateaued ever since the initial boon; therefore, farmers are constantly making decisions between increasing profits, investing in more environmentally friendly farming practices, or improving the welfare of farm workers, and any decision pits one pillar against the other. Research in sustainability should endeavor to ensure that it should not be an either/or situation between the three pillars of sustainability, and that even though it may appear as such, one pillar does not have to be sacrificed for the other. As shown in Figure 6, research should endeavor to make sure that the aim of sustainability is less to achieve all three pillars, but more to optimally balance and reconcile all three pillars relative to the resources of the farm and in the prevailing context of the country. The South African case study has shown that any effort at any time to place one pillar ahead of another, for any reason, belies the overall sustainability of the farm, and research in sustainability should make this clear.

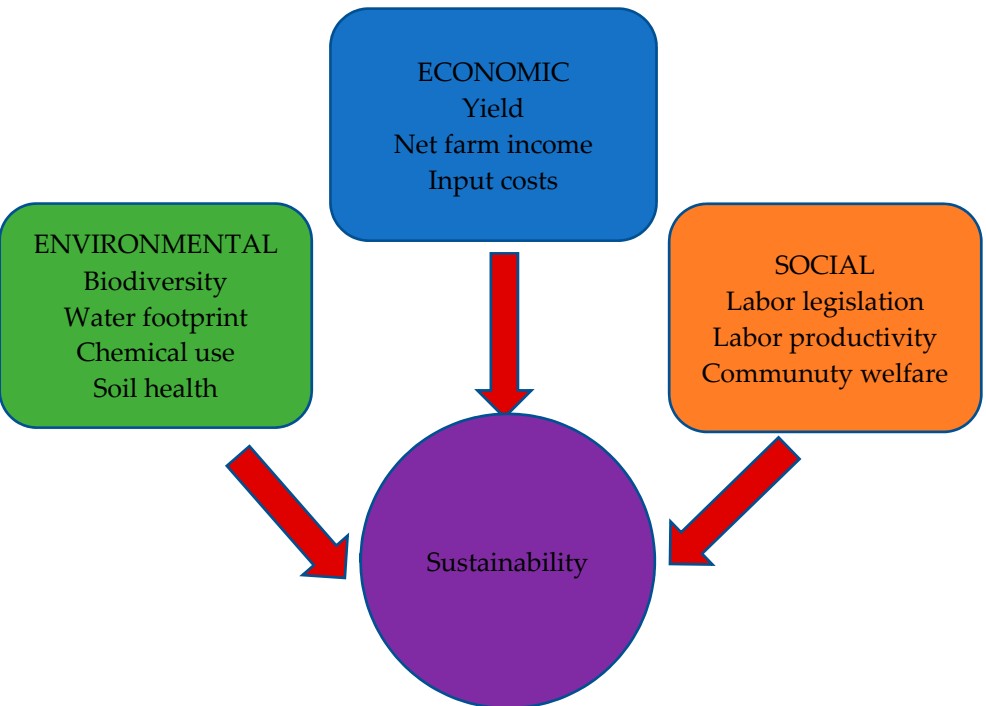

**Figure 6.** Graph showing the indicators of sustainability required to achieve overall sustainable grape and wine production.

Furthermore, even though there are a variety of assessment methods for the three pillars of sustainability, a comprehensive sustainability assessment of all three pillars simultaneously for the partial or entire value chain of grapes and wine are absent and sorely needed. Part of the reason for this lack of research is due to the paucity of the amount of data needed, especially for off-farm activities. Additionally, data for on-farm activities, especially for the non-productive stage of the grapevine, are not always available. This is disconcerting because it has been argued that a lack of quantitative data makes it difficult and even impossible to see and assess opportunities for improving performance and monitoring progress towards the end goal of sustainability [15]. Another reason is the lack of measurable context-specific indicators for economic, environmental, and social indicators for the South African grape and wine industry that would usually precede any sort of sustainability assessment.

In terms of the three pillars, considerable research gaps still exist. Firstly, in terms of economic sustainability for grape and wine production in the context of climate change, even though there theoretically exists a point where further increases in temperature will depress grape and wine prices, in practice, this point is not known [178]. Future research should aim to link increases in temperature with grape prices to understand where climate change starts being detrimental to grape and wine production, especially for warm category regions such as South Africa.

Regarding environmental sustainability amidst grape and wine production, only a handful of "noble" grape varieties are planted worldwide, relegating the other considerable numbers of varieties to very little limited hectares. This needs to be remedied, because many of these local or indigenous varieties may very well play a significant role in the future in the context of a warming climate; these "neglected" varieties could well be adapted to extreme and harsh climate due to years of "neglect". However, consumer acceptance of these varieties needs to be investigated simultaneously [179]. Furthermore, there is limited research on environmental problems that are very important and informative to farm managers. For example, because as detrimental as soil erosion is in all forms of crop production, research on it is still limited [180–182].

In the context of social sustainability in climate change, there is a need to assess the effectiveness of schemes such as the Wine and Agricultural Ethical Trade Association (WIETA), Fair Trade South Africa, and Sustainability Initiative of South Africa (SIZA) exclusively from workers' perspectives, because the effectiveness of these schemes are unconfirmed and largely up for debate; with the increased dominance of these schemes by retailers, the farm workers who they are supposed to support are ironically being left out of the conversation [109,112,113,183].

## 5. Conclusions

Sustainability has become a catch-all phrase for practically all efforts to remedy the detrimental impacts of conventional agriculture, even in the grape and wine industry that traditionally has not been viewed as a particularly environmentally impactful industry. The historical context of South Africa shows that sustainability amidst climate change is very important to the grape and wine industry, especially for reasons of environmental stewardship, higher profits, and stakeholders' pressure. Research has shown that table grape farms are more economically sustainable than wine farms, but the climate change effects on profitability is unpredictable. In addition to the inefficient use of water and chemicals, soil erosion, pest, diseases, and irrigation pressure are bound to intensify as the climate warms. Regardless of the various efforts to improve the welfare of farm workers, social sustainability at the level of the farm leaves a lot to be desired and this has no sign of changing anytime soon. However, there are various short-term (changes in viticultural practices, soil management practices and integrated pest management) and long-term (changes in training systems, changes in grape and rootstock varieties and vineyard relocation) adaptation measures to mitigate against the current and potential impacts of climate change in viticulture and winemaking, but these face barriers in adoption.

**Author Contributions:** Conceptualization, O.G. and E.B.; methodology, O.G.; formal analysis, O.G.; resources, O.G., S.K. and E.B.; investigation, O.G.; writing—original draft preparation, O.G.; writing—review and editing, O.G., S.K., and E.B.; visualization, S.K. and E.B.; supervision, S.K. and E.B. All authors have read and agreed to the published version of the manuscript.

**Funding:** This research received no external funding.

**Data Availability Statement:** No new data were created or analyzed in this study.

**Acknowledgments:** The authors would like to thank Elsje Dippenaar from the Sustainable Agriculture Masters programme, Stellenbosch University, for her professional networking that laid the groundwork and that was instrumental in making this paper come to fruition.

**Conflicts of Interest:** The authors declare no conflict of interest.

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
