# Peer review of "The 3Ps (Profit, Planet, and People) of Sustainability amidst Climate Change: A South African Grape and Wine Perspective"

_sustainability, doi:10.3390/su13052910_

Round 1
Reviewer 1 Report
The authors have well worked out previous comments and I have no more comments.
Author Response
Dear editor,
With this I wish to submit my revised version of the manuscript No. 1102987.
RESPONSE TO REVIEWERS’ COMMENTS ON OUR PAPER: THE 3Ps (PROFIT, PLANET AND PEOPLE) OF SUSTAINABILITY AMIDST CLIMATE CHANGE: A SOUTH AFRICAN GRAPE AND WINE PERSPECTIVE.
First of all, we would like to thank the reviewer for his/her time and effort to improve our paper and for the compliments they made about our paper. We tried to answer and address all the comments and answers that were posed by the reviewers. The corrections are also in the submitted manuscript (in the colour red).
To address all the issues efficiently we copied the reviewer comments below (in bold italics). and answered each issue below.
On behalf of my co-authors,
Omamuyovwi Gbejewoh
Reviewer 2
Comment: Paragraph 4.2 may represent an introduction to the South African context instead of a result.
Response: At first glance that may be the case. However, the authors were not acutely aware what the historical and political context of grape and wine production meant for sustainability. It was during the review process the authors became aware of the real meaning of the historical and political context of the country and more importantly, what this meant for understanding the pillars of sustainability in the country. Furthermore, almost all papers that dealt with sustainability (especially social sustainability) in South Africa gave a brief historical and political context of the country as a preamble to understanding (this was stressed) the sustainability results presented in these papers, thus analyzing its importance as much a part of the results. (See Kritzinger et al., 2004; McEwan and Bek, 2006;2009; Linton, 2012; Devereux, 2020, just to list a few). For good measure, the historical and political context of the country was part of the review process search results.
Comment: How may a reader find a structured answer to the second objective? The suggestion that that the research questions were the starting point to prepare the discussions coherently. Nevertheless, the paragraphs from 4.3 to 4.14 seems more like a list of topics unless a general framework is presented before.
Response: In response to the above comment, a framework was provided for a structured answer to the second objective. This is given below
“The framework for the analysis for the current trends in sustainability will be to discuss each pillar (economic, environmental, and social) of sustainability separately at first as a standalone concept. Thereafter, climate change will be introduced into these pillars and thus, the effects of climate change in each pillar (economic, environmental, and social) of sustainability will be further discussed separately” (Line 73-78)
Comment: Authors should declare the string used to select the papers to allow researchers to repeat the analysis
Response: The search strings were provided as below.
“The search string words in Web of Science and Scopus Database were: TITLE-ABS-KEY ((“sustainability*” OR “sustainability pillar*” OR “climate change*”) AND (“viticulture*” OR “vineyards*” OR “wine*” OR “grape*”))” (Line 94-96).
Reviewer 3
Comment: This figure can be improved too basic. Use different color, font... make attractive.
Response: Figure 1 was improved per the reviewer’s suggestion. (Line 106-108).
Comment: The graphs are poor. They need to be updated
Response: Figure 3 was updated. (Line 501-504)
Comment: You should highlight that this is not unique in the vineyards, other crops shows similar problems.
Response: This was done as the per the comment above. See below
“It should be noted though that extensive soil loss is not limited to vineyards as different authors have reported similar problems in various other crops”. (Line 364-365).
Comment: I suggest here that one of the lack of the current management is the lack of research that will inform the managers of the environmental problems Here some examples, soil erosion is a problem and we have less than 100 research papers in the world.
Response: This research gap was added to the manuscript as below
“Furthermore, there is limited research on environmental problems that are very important and informative to farm managers. For example, as much of a nuisance as soil erosion is in all forms of crop production, research on it is still limited” (Line 551-554).

Reviewer 2 Report
The authors improve the paper, but the organisation of the paper is not yet fully satisfactory.
In previuos revision I posed to the authors the following comment:
Comment: With regard to the objectives, the paper does not clarify in the introduction which research questions the review would like to answer...
Response: This was corrected and the research questions the paper hoped to answer is now stated as research objectives in the introduction.
Replay: the author introduces two objectives:"(i) analyse why sustainability is important to grape and wine farmers (ii) analysing current trends in the economic, environmental, and social sustainability of grape and wine production and how climate change is affecting these trends". How a reader can find a structured answer to these objectives in the paper? I can read paragraph 4.1 that justify the first question. But, what about the rest of the paper?
Paragraph 4.2 may represent an introduction to the South African context instead of a result.
The suggestion to present the research questions was the starting point to prepare the discussion coherently. Nevertheless, the paragraphs from 4.3 to 4.14 seems more a list of topics unless a general framework is presented before.
Minor comments:
Authors should declare the string used to select the papers to allow researchers to repeat the analysis.
Author Response
Dear editor,
With this I wish to submit my revised version of the manuscript No. 1102987.
RESPONSE TO REVIEWERS’ COMMENTS ON OUR PAPER: THE 3Ps (PROFIT, PLANET AND PEOPLE) OF SUSTAINABILITY AMIDST CLIMATE CHANGE: A SOUHT AFRICAN GRAPE AND WINE PERSPECTIVE.
First of all, we would like to thank the reviewer for his/her time and effort to improve our paper and for the compliments they made about our paper. We tried to answer and address all the comments and answers that were posed by the reviewers. The corrections are also in the submitted manuscript (in the colour red).
To address all the issues efficiently we copied the reviewer comments below (in bold italics). and answered each issue below.
On behalf of my co-authors,
Omamuyovwi Gbejewoh
Reviewer 2
Comment: Paragraph 4.2 may represent an introduction to the South African context instead of a result.
Response: At first glance that may be the case. However, the authors were not acutely aware what the historical and political context of grape and wine production meant for sustainability. It was during the review process the authors became aware of the real meaning of the historical and political context of the country and more importantly, what this meant for understanding the pillars of sustainability in the country. Furthermore, almost all papers that dealt with sustainability (especially social sustainability) in South Africa gave a brief historical and political context of the country as a preamble to understanding (this was stressed) the sustainability results presented in these papers, thus analyzing its importance as much a part of the results. (See Kritzinger et al., 2004; McEwan and Bek, 2006;2009; Linton, 2012; Devereux, 2020, just to list a few). For good measure, the historical and political context of the country was part of the review process search results.
Comment: How may a reader find a structured answer to the second objective? The suggestion that that the research questions were the starting point to prepare the discussions coherently. Nevertheless, the paragraphs from 4.3 to 4.14 seems more like a list of topics unless a general framework is presented before.
Response: In response to the above comment, a framework was provided for a structured answer to the second objective. This is given below
“The framework for the analysis for the current trends in sustainability will be to discuss each pillar (economic, environmental, and social) of sustainability separately at first as a standalone concept. Thereafter, climate change will be introduced into these pillars and thus, the effects of climate change in each pillar (economic, environmental, and social) of sustainability will be further discussed separately” (Line 73-78)
Comment: Authors should declare the string used to select the papers to allow researchers to repeat the analysis
Response: The search strings were provided as below.
“The search string words in Web of Science and Scopus Database were: TITLE-ABS-KEY ((“sustainability*” OR “sustainability pillar*” OR “climate change*”) AND (“viticulture*” OR “vineyards*” OR “wine*” OR “grape*”))” (Line 94-96).
Reviewer 3
Comment: This figure can be improved too basic. Use different color, font... make attractive.
Response: Figure 1 was improved per the reviewer’s suggestion. (Line 106-108).
Comment: The graphs are poor. They need to be updated
Response: Figure 3 was updated. (Line 501-504)
Comment: You should highlight that this is not unique in the vineyards, other crops shows similar problems.
Response: This was done as the per the comment above. See below
“It should be noted though that extensive soil loss is not limited to vineyards as different authors have reported similar problems in various other crops”. (Line 364-365).
Comment: I suggest here that one of the lack of the current management is the lack of research that will inform the managers of the environmental problems Here some examples, soil erosion is a problem and we have less than 100 research papers in the world.
Response: This research gap was added to the manuscript as below
“Furthermore, there is limited research on environmental problems that are very important and informative to farm managers. For example, as much of a nuisance as soil erosion is in all forms of crop production, research on it is still limited” (Line 551-554).

Reviewer 3 Report
The paper has too many weak points
The graphs should be improved
And there is a lack of key literature
In general, is a good paper
see comments attached

Author Response
Dear editor,
With this I wish to submit my revised version of the manuscript No. 1102987.
RESPONSE TO REVIEWERS’ COMMENTS ON OUR PAPER: THE 3Ps (PROFIT, PLANET AND PEOPLE) OF SUSTAINABILITY AMIDST CLIMATE CHANGE: A SOUHT AFRICAN GRAPE AND WINE PERSPECTIVE.
First of all, we would like to thank the reviewer for his/her time and effort to improve our paper and for the compliments they made about our paper. We tried to answer and address all the comments and answers that were posed by the reviewers. The corrections are also in the submitted manuscript (in the colour red).
To address all the issues efficiently we copied the reviewer comments below (in bold italics). and answered each issue below.
On behalf of my co-authors,
Omamuyovwi Gbejewoh
Reviewer 2
Comment: Paragraph 4.2 may represent an introduction to the South African context instead of a result.
Response: At first glance that may be the case. However, the authors were not acutely aware what the historical and political context of grape and wine production meant for sustainability. It was during the review process the authors became aware of the real meaning of the historical and political context of the country and more importantly, what this meant for understanding the pillars of sustainability in the country. Furthermore, almost all papers that dealt with sustainability (especially social sustainability) in South Africa gave a brief historical and political context of the country as a preamble to understanding (this was stressed) the sustainability results presented in these papers, thus analyzing its importance as much a part of the results. (See Kritzinger et al., 2004; McEwan and Bek, 2006;2009; Linton, 2012; Devereux, 2020, just to list a few). For good measure, the historical and political context of the country was part of the review process search results.
Comment: How may a reader find a structured answer to the second objective? The suggestion that that the research questions were the starting point to prepare the discussions coherently. Nevertheless, the paragraphs from 4.3 to 4.14 seems more like a list of topics unless a general framework is presented before.
Response: In response to the above comment, a framework was provided for a structured answer to the second objective. This is given below
“The framework for the analysis for the current trends in sustainability will be to discuss each pillar (economic, environmental, and social) of sustainability separately at first as a standalone concept. Thereafter, climate change will be introduced into these pillars and thus, the effects of climate change in each pillar (economic, environmental, and social) of sustainability will be further discussed separately” (Line 73-78)
Comment: Authors should declare the string used to select the papers to allow researchers to repeat the analysis
Response: The search strings were provided as below.
“The search string words in Web of Science and Scopus Database were: TITLE-ABS-KEY ((“sustainability*” OR “sustainability pillar*” OR “climate change*”) AND (“viticulture*” OR “vineyards*” OR “wine*” OR “grape*”))” (Line 94-96).
Reviewer 3
Comment: This figure can be improved too basic. Use different color, font... make attractive.
Response: Figure 1 was improved per the reviewer’s suggestion. (Line 106-108).
Comment: The graphs are poor. They need to be updated
Response: Figure 3 was updated. (Line 501-504)
Comment: You should highlight that this is not unique in the vineyards, other crops shows similar problems.
Response: This was done as the per the comment above. See below
“It should be noted though that extensive soil loss is not limited to vineyards as different authors have reported similar problems in various other crops”. (Line 364-365).
Comment: I suggest here that one of the lack of the current management is the lack of research that will inform the managers of the environmental problems Here some examples, soil erosion is a problem and we have less than 100 research papers in the world.
Response: This research gap was added to the manuscript as below
“Furthermore, there is limited research on environmental problems that are very important and informative to farm managers. For example, as much of a nuisance as soil erosion is in all forms of crop production, research on it is still limited” (Line 551-554).

This manuscript is a resubmission of an earlier submission. The following is a list of the peer review reports and author responses from that submission.
Round 1
Reviewer 1 Report
This study is well written, with a clear resarch question: reviewing current state of knowledge with regards to the three pillars of sustainability and with regards to climate change in the grape&wine sector in South Africa.
The authors use a "state-of-the-arts" methodology, following the guidelines set by PRISMA, in order to perform a structured review. This is very useful for the selection of papers.
The presentation of the results of the review, however, is too generic, in my view. It mixes and broadens many concepts that could be easily summarized. Indeed, the main point of the research is that there are no siginificant studies on the topic.
The presentation of the analysis should be more rigourous. The authors should/could use descriptive statistics to make the paper more scientifically sound. Using simple frequency computations and/or cross tabulations between key indicators may help improving the scientific soundness of the work. For instance the author should present the frequencies of what studies contain/does not contain the selected issue. Graphs and tables could help in presenting results and making the work more organized and less "generic".
Another point refers to understanding how to use the results. What can be done to improve sustainability and studies on sustainability? How can the South African case can represent a key case-study/example for other countries?
In other words, what is the added value, to research and scientific knowledge?
Reviewer 2 Report
The document aims to offer a systematic review of the state of the art regarding the sustainability assessment of viticulture in South Africa, in the context of clmate change.
The authors address a central issue with respect to the journal's interests, however, it has several limitations regarding the definition of objectives, the appropriateness of applying a systematic review as well as the way in which it was conducted.
The title is not coherent with the content of the paper that marginally discuss South African grape production.
With regard to the objectives, the paper does not clarify in the introduction which research questions the review would like to answer, limiting to say that there is no holistic analysis of the three pillars of sustainability in South Africa in the context of climate change, intending to bridge this gap. However, the research concludes on page 9 line 430-431 by saying that research on the three pillars of sustainability in the context of climate change is practically non-existent.
On the basis of these conclusions, supported by the results presented in the paper which refer to research conducted mainly in contexts other than South Africa, a doubt arises as to the usefulness of a systematic review: the method should offer the opportunity to give an answer to research questions for which there is a rich availability of research results.
Finally, the PRISMA methodology is not set up sufficiently as it is incomplete in the presentation of the paper selection procedure, product quality control and the way the results are discussed. We recommend reading the following papers for reference:
- from Costa Maynard, D., Vidigal, M. D., Farage, P., Zandonadi, R. P., Nakano, E. Y., & Botelho, R. B. A. (2020). Environmental, social and economic sustainability indicators applied to food services: a systematic review. Sustainability, 12 (5), 1804.
- Sesini, G., Castiglioni, C., & Lozza, E. (2020). New trends and patterns in sustainable consumption: A systematic review and research agenda. Sustainability, 12 (15), 5935.
The authors should previously build a framework for the discussion of the topic in the introduction. Furthermore, it may be useful develop a framework for discussing together the three pillar of sustainability together in the context of climate change. The results session should involve only selected papers.
Reviewer 3 Report
Dear author
I found the paper excellent. This is a great contribution.
The issue of sustainability is very discussed and you clarify all the key aspects of the definition
I fully support the publication of this paper that brings relevant information to all the readers
I was deeply reviewing the paper and the comments are attached and are related to the last publication you probably could not found as they are very recent publications
See that some recent publication of Rodrigo are relevant to show the update in your literature review
Sincerely